# Ability of a Polyphenol-Rich Nutraceutical to Reduce Central Nervous System Lipid Peroxidation by Analysis of Oxylipins in Urine: A Randomized, Double-Blind, Placebo-Controlled Clinical Trial

**DOI:** 10.3390/antiox12030721

**Published:** 2023-03-14

**Authors:** Raúl Arcusa, Juan Ángel Carillo, Begoña Cerdá, Thierry Durand, Ángel Gil-Izquierdo, Sonia Medina, Jean-Marie Galano, María Pilar Zafrilla, Javier Marhuenda

**Affiliations:** 1Faculty of Health Sciences, Universidad Católica de San Antonio, 30107 Murcia, Spain; rarcusa@ucam.edu (R.A.); jacarrillo4@alu.ucam.edu (J.Á.C.); bcerda@ucam.edu (B.C.); jmarhuenda@ucam.edu (J.M.); 2Institut des Biomolécules Max Mousseron (IBMM), Pôle Chimie Balard Recherche, UMR 5247, CNRS, Université de Montpellier, ENSCM 1919 Route de Mende, CEDEX 05, 34293 Montpellier, France; thierry.durand@umontpellier.fr (T.D.); jgalano@univ-montp1.fr (J.-M.G.); 3Research Group on Quality Safety and Bioactivity of Plant Foods, Food Science and Technology Department, CEBAS-CSIC, 30100 Murcia, Spain; angelgil@cebas.csic.es (Á.G.-I.); smescudero@cebas.csic.es (S.M.)

**Keywords:** lipid peroxidation, oxylipins, polyphenols

## Abstract

Isoprostanes (IsoPs) are lipid peroxidation biomarkers that reveal the oxidative status of the organism without specifying which organs or tissues it occurs in. Similar compounds have recently been identified that can assess central nervous system (CNS) lipid peroxidation status, usually oxidated by reactive oxygen species. These compounds are the neuroprostanes (NeuroPs) derived from eicosapentaenoic acid (EPA) and docosahexaenoic acid (DHA) and the F_2t_-dihomo-isoprotanes derived from adrenic acid (AdA). The aim of the present investigation was to evaluate whether the long-term nutraceutical consumption of high polyphenolic contents (600 mg) from fruits (such as berries) and vegetables shows efficacy against CNS lipid peroxidation in urine biomarkers. A total of 92 subjects (47 females, 45 males, age 34 ± 11 years old, weight 73.10 ± 14.29 kg, height 1.72 ± 9 cm, body mass index (BMI) 24.40 ± 3.43 kg/m^2^) completed a randomized, cross-over, double-blind study after an intervention of two periods of 16 weeks consuming either extract (EXT) or placebo (PLA) separated by a 4-week washout period. The results showed significant reductions in three AdA-derived metabolites, namely, 17-*epi*-17-F_2t_-dihomo-IsoPs (Δ −1.65 ng/mL; *p* < 0.001), 17-F_2t_-dihomo-IsoPs (Δ −0.17 ng/mL; *p* < 0.015), and *ent*-7(RS)-7-F_2t_-dihomo-IsoPs (Δ −1.97 ng/mL; *p* < 0.001), and one DHA-derived metabolite, namely, 4-F_4t_-NeuroP (Δ −7.94 ng/mL; *p* < 0.001), after EXT consumption, which was not observed after PLA consumption. These data seem to show the effectiveness of the extract for preventing CNS lipid peroxidation, as determined by measurements of oxylipins in urine through Ultra-High-Performance Liquid Chromatography triple quadrupole tandem mass spectrometry (UHPLC-QqQ-ESI-MS/MS).

## 1. Introduction

Oxidative stress (OS) is known as an imbalance between oxidizing and anti-oxidizing agents, affecting the physiological function of the organism [1] that could cause damage to lipids, proteins, and DNA structures, among others [2]. This imbalance could be caused by a low intake of antioxidants, depletion of endogenous antioxidants, or an increase in reactive species [3]. Whilst it is known that the organism synthesizes reactive oxygen species (ROS) and nitrogen species (NOS) in small enough quantities that it is capable of neutralizing, when this production is excessive and prolonged over time, it can cause damage to cell structures and functions and may lead to irreparable damage [4]. Lipid peroxidation of membranes can lead to impaired functions, inactivation of its receptors and enzymes, increased permeability to ions, decreased fluidity, and, ultimately, rupture [5].

Several problems arise in the assessment of OS, particularly in the determination of scientifically validated biomarkers [6]. In addition, for such biomarkers to be useful, they must meet certain criteria, such as showing specificity for a given disease, having prognostic value, correlating with disease activity, remaining reasonably stable, being easily accessible in tissues, and are cost-effective to measure on a large scale [1]. Despite the fact that it is common to assess the total antioxidant capacity in subjects’ blood plasma, this procedure may not provide accurate information on the organism’s state [7]. The scientific literature has described how numerous phytochemicals are fleetingly metabolized into molecules with altered biological properties [7]. In addition, it is of greater interest to evaluate the activity of the organism’s antioxidant enzymes [8], which support the main antioxidant defense load of the organism [9].

There are different pathways involved that explain the antioxidant capacity of polyphenols, highlighting the elimination of ROS, the attenuation in the synthesis of these radicals by inhibiting enzymes involved in their production, and the increase in endogenous antioxidant defenses [10]. Moreover, thanks to their structure and the presence of hydroxyl groups attached to the aromatic ring, they exert antioxidant activity to neutralize unpaired electrons from free radicals (FR), donating hydrogens, and chelating metallic ions [11]. For flavonoids, the hydroxyl of the B-ring appears to be responsible for scavenging ROS and NOS by hydrogen and electron donation to hydroxyl, peroxyl, and peroxynitrite radicals, giving rise to stable flavonoid radicals [12,13].

Oxylipins are oxygenated bioactive lipids [14], synthesized as metabolites from the oxidation of polyunsaturated fatty acids (PUFAs). The scientific literature describes how altered oxylipin signaling is associated with different types of cardiovascular diseases (CVDs) such as diabetes, hypertension, hemostasis, thrombosis, and hyperlipidemia [15]. Oxylipins originate after damage or stimulus; due to their short lifespan they are not stored, but are synthesized de novo in a tightly regulated manner [16]. Linoleic acid ((LA), omega-6 family) and alpha-linolenic acid ((ALA), omega-3 family) are the main precursors of oxylipins. In the LA metabolism pathway, dihomo-gamma-linolenic acid (DGLA) and arachidonic acid (AA) are synthesized, while in the ALA pathway, EPA and DHA are generated [17,18]. 

The different types of oxylipins, including IsoPs, were discovered by Morrow and collaborators in the 1990s [19]. IsoPs are prostaglandin-like eicosanoids, synthesized by a non-enzymatic pathway independent of cyclooxygenase (COX), via ROS-induced AA peroxidation [20,21]. IsoPs are fleetingly metabolized and excreted in urine, and their quantification is useful to assess the oxidative status at a particular time [22,23,24].

One of the limitations of IsoPs is that, despite revealing the oxidative status of the organism, it does not specify which organs or tissues it occurs in, unless the IsoP levels in the cerebrospinal fluid are directly assessed [25]. Nevertheless, recent compounds have been described as biomarkers of CNS lipid peroxidation, such as NeuroPs originating from EPA and/or DHA (which produce F_3_-IsoPs and F_4_-IsoPs, respectively) and F_2t_-dihomo-IsoPs, derived from AdA [26,27,28], as illustrated in Figure 1. These compounds can be assessed in different biological fluids, with plasma and urine being the methods of choice [29,30,31,32].

AA is evenly distributed in the brains’ gray and white matter, as well as inside glia and neurons [33]. Urine assessment of NeuroPs and F_2t_-dihomo-IsoP, which are both present in brain tissues, could be an additional and more reliable indicator of neuronal oxidative and myelin sheath damage, respectively, because DHA is abundant in the gray matter of the brain and AdA is abundant in the white matter [30,31]. 

Since the CNS is prone to ROS production and lacks an adequate antioxidant system, a constant increase in ROS and NOS can occur [34]. High oxygen consumption by the brain results in excessive ROS production since neuronal membranes are rich in PUFAs and therefore have increased vulnerability to FR attack to its high number of double bonds [35]. 

In this context, the aim of the present investigation was to evaluate whether the consumption of nutraceutical-based fruits (such as berries) and vegetables with a high polyphenol content was effective to prevent CNS lipid peroxidation by attenuating urine metabolites derived from EPA, DHA, and AdA.

## 2. Materials and Methods

### 2.1. Chemical and Reagent

Three NeuroPs (4(RS)-4F_4t_-NeuroP, 4-*epi*-4-F_3t_-NeuroP, 4-F_4t_-NeuroP) and three F_2t_-dihomo-IsoPss (17-*epi*-17-F_2t_-dihomo-IsoP, 17-F_2t_-dihomo-IsoP, *Ent*-7(RS)-7F_2t_-dihomo-IsoP) were synthesized by Durand’s team at the Institut des Biomolecules Max Mosseron (IBMM) (Montpellier, France). Table 1 shows the different NeuroPs and F_2t_-dihomo-IsPs molecules tested in this study.

The enzyme β-glucuronidase, type H2 from Helix pomatia and BIS-TRIS (bis-(2hydroxyethyl)-amino-tris(hydroxymethyl)-tris(hydroxymethyl)-methane) were obtained from Sigma-Aldrich (St Louis, MO, USA). All LC-MS-grade solvents were obtained from J.T. Baker (Phillipsburg, NJ, USA). Hydrochloric acid, hexane, trichloroacetic acid, and ethyl acetate were obtained from Panreac (Caste3llar del Vallés, Barcelona, Spain). Strata X-AW solid-phase extraction (SPE) cartridges, 100 mg per 3 mL, were purchased from Phenomenex (Torrance, CA, USA). Water was treated in a Milli-Q water purification system from Millipore (Bedford, MA, USA).

### 2.2. Clinical Trial Design 

The study consisted of a randomized, double-blind, crossover, two-arm, placebo-controlled, sex-stratified, single-center clinical trial, which was described in detail and illustrated in a previous publication [36]. The intervention phase lasted for 36 weeks, divided into two phases of 16 weeks wherein subjects consumed either extract or placebo, separated by a 4-week washout period, after which subjects transitioned to the opposite arm of the study. 

During the intervention, the subjects visited the laboratory four times, corresponding to the beginning and end of each 16-week phase. During these visits, in addition to checking adherence to treatment, urine was collected in the 24 h prior to the visit. Researchers recorded the total volume of urine and stored the different samples at −80 °C for subsequent analysis.

Prior to starting the intervention, the protocol was approved by the Institutional Ethical Committee Board of the Catholic University San Antonio of Murcia (UCAM), dated 24 November 2017, under the code: CE111072.

### 2.3. Participants

The subjects had to meet certain inclusion criteria: signing an informed consent form, not consuming more than three servings of fruit and vegetables per day, having a BMI between 18.5 and 35 kg/m^2^, and being aged between 18 and 65 years old. At the same time, the fulfillment of only one exclusion criterion was sufficient reason for not being a candidate for the study. These criteria include: changes in physical activity habits during the intervention; being pregnant or breastfeeding; being smokers; taking any medication or food supplements, nutraceuticals, multivitamins, etc.; being on a weight-loss, vegan, or vegetarian diet; typically consuming more than three glasses of alcohol (wine, beer) per day; having sleep problems; giving a blood donation (0.5 L) within the last month; and having undergone major surgery in the last three months. 

### 2.4. Test Supplement

The product under investigation is a nutraceutical consisting of a mixture of 36 different sources of berries, vegetables, and dried fruits, which are explained in detail a previous publication [37]. In order to blind both researchers and subjects, a PLA of similar visual appearance was manufactured and provided by the same company. The daily dose consisted of six capsules, taken twice a day (three capsules in the morning and three in the afternoon).

Bresciani and collaborators previously characterize the product through UHPLC-QqQ-MS, and showed that this daily dose of the product offered a total of 600 mg of polyphenols, detecting a total of 119 different phenolic compounds [38]. The same research group performed a bioavailability study showing that of the 92 molecules monitored, 20 of them could be detected in the plasma of the subjects, all of them in the form of conjugates and at different times depending on the site of absorption from the gastrointestinal tract [39]. 

### 2.5. Extraction of Human Oxylipins in Urine Samples

The extraction of NeuroPs and F_2t_-dihomo-IsoPs was performed by solid-phase extraction as described in previous works [22,40,41,42] and whose methodology is detailed in [36].

### 2.6. UHPLC-QqQ-MS/MS Analysis of Oxylipins

NeuroPs and F_2t_-dihomo-IsoPs analysis was performed by chromatographic separation using UHPLC-QqQ-ESI-MS/MS following a previously published methodology [22,40,41,42]. The protocol is fully described elsewhere [36], employing a different type of column (C18 column (2.1 × 50 mm, pore size of 1.7 μm)) (Waters, MA, USA). 

### 2.7. Statistical Analysis

Descriptive analysis was performed (mean and standard deviation (SD)) on all the quantitative study variables, for baseline conditions and in their evolution, using the Kolmogorov–Smirnov test to verify the normal distribution of the continuous data, both in the EXT and PLA consumption periods. At baseline, Student’s t-distribution comparisons were performed between the two arms of the study to verify if the groups were homogeneous. To analyze the evolution of the variables between groups, an analysis of variance for repeated measures was performed with an intrasubject factor (time manner: baseline and final for each study arm) and an intersubject factor (product: experimental and placebo). For the post hoc analysis, the Bonferroni test was used. Comparisons were made for those effects that were significant with the option of assuming or not assuming equality of variances. In the set of statistical tests, the significance level used was 0.05, and the statistical analysis was performed using SPSS Statistics 27 (SPSS, Inc., Chicago, IL, USA).

## 3. Results

### 3.1. Study Population

As summarized in the flow diagram in Figure 2, after the recruitment and selection phase, a total of 117 subjects of both sexes were selected after verification of the inclusion and exclusion criteria, and they were randomly assigned to one of the two groups. Finally, 108 subjects were distributed across two homogeneous groups and started the intervention (Table 2). During this phase, and after losses due to lack of follow-up and dropouts, a total of 92 volunteers completed the treatment.

### 3.2. Oxylipins

Urine samples were collected at the beginning and end of the intervention and analyzed to evaluate lipid peroxidation in the CNS, through oxylipins derived from EPA, DHA, and AdA. Of the six oxylipins analyzed, we were able to quantify four: three derived from AdA and one from DHA, whose values are shown in Table 3.

#### 3.2.1. Oxylipins Derived from Adrenic Acid

-17-*epi*-17-F_2t_-dihomo-IsoP

As shown in Figure 3, the 17-*epi*-17-F_2t_-dihomo-IsoP values were non-significantly reduced after the consumption of PLA (Δ −0.121), while after the consumption of EXT, they were reduced in a more pronounced and significant way (Δ −1.65 ng/mL). Furthermore, it is noteworthy that after comparing the evolution between groups at the end of the intervention, statistically significant differences were observed in the comparison of the final timepoint between PLA and EXT, which seems to confirm that the consumption of EXT may be effective in reducing 17-*epi*-17-F_2t_-dihomo-IsoP levels.

-17-F_2t_-dihomo-IsoP

As shown in Figure 4, the values of 17-F_2t_-dihomo-IsoP remained stable after consumption of PLA (Δ 0), whereas after the consumption of EXT, they were significantly reduced (Δ −0.17 ng/mL). However, comparing the evolution between treatment groups at the end of the intervention, no statistically significant differences were observed. These data suggest that the consumption of EXT reduces 17-F_2t_-dihomo-IsoP levels compared to PLA.

-*ent*-7(RS)-7-F_2t_-dihomo-IsoP

As shown in Figure 5, the *ent*-7(RS)-7-F_2t_-dihomo-IsoP values started from disparate and non-homogeneous values at the beginning of the intervention, with higher values in the EXT group. Comparing the evolution in each group separately, it was observed that *ent*-7(RS)-7-F_2t_-dihomo-IsoP levels were not altered after the consumption of PLA (Δ 0.03), whereas after the consumption of EXT, they were significantly reduced (Δ −1.97 ng/mL). After comparing the evolution between the groups at the end of the intervention, statistically significant differences were observed, and when the comparison was made at the end of the study, significant differences from baseline were no longer observed. These results seem to show that EXT is effective in reducing *ent*-7(RS)-7-F_2t_-dihomo-IsoP levels. However, given that at the beginning of the intervention, the values were very variable, and the reason is yet known, caution should be exercised in interpreting these results.

#### 3.2.2. Oxylipins Derived from Docosahexaenoic Acid

The only NeuroP quantified was 4-F_4t_-NeuroP, the most representative NeuroP (Figure 6), whose values did not vary significantly (Δ 0.38) after the consumption of PLA, while after the consumption of EXT, they were significantly reduced (Δ −7.94 ng/mL). Comparing the evolution between the groups at the end of the intervention, statistically significant differences were observed. Furthermore, to increase the statistical power size effect, the difference at the end of the intervention between PLA and EXT were analyzed, and statistically significant differences were also seen. Therefore, the consumption of EXT seems to be effective in reducing 4-F_4t_-NeuroP levels.

## 4. Discussion

The CNS is highly vulnerable to ROS-mediated injury due to the brain’s high oxygen consumption (given its high energy demands), high PUFA levels, and weak antioxidant defenses [26], leading to the overproduction of ROS [35]. Specifically, the brain uses more than 20% of all oxygen consumed by mitochondrial respiration [43].

The brain demands high amounts of oxygen, which makes it prone to OS, with oxidative brain injury being a major cause of neurological disorders such as epilepsy [44]. DHA oxidation in the CNS is associated with various neurodegenerative disorders, including Rett syndrome, amyotrophic lateral sclerosis, Huntington’s disease, Parkinson’s disease, and Alzheimer’s disease [45,46].

In the present investigation, we considered evaluating three NeuroPs corresponding to series 4 and three F_2t_-dihomo-IsoPs corresponding to series 7 and 17 (Table 1) to assess DHA and AdA oxidation, respectively. Of the six compounds analyzed, four were quantified, one NeuroP and three F_2t_-dihomo-IsoPs, the results of which are shown in Table 3.

In the case of the only NeuroP quantified, 4-F_4t_-NeuroP, a significant reduction in its values was observed only after the consumption of EXT, which did not occur after the consumption of PLA. Furthermore, observing the differences at the end of the intervention between groups and comparing their evolution, significant differences were also observed, which reinforces the statistical power of the results.

Within the F_4t_-NeuroPs, 4-F_4t_-NeuroP and 10-F_4t_-NeuroP are the most frequently reported, being elevated in the plasma in a multitude of neurological diseases [47] and cognitive impairment disorders [48], as well as in the plasma and urine of smokers [49] and type 2 diabetic patients [50]. Signorini et al. conducted an investigation to evaluate the clinical relevance of 4-F_4t_-NeuroP and 10-F_4t_-NeuroP in four neurological diseases (Rett syndrome, Down syndrome, multiple sclerosis, and autism spectrum disorder) compared with a control group of age-matched subjects, and found that the levels of 10-F_4t_-NeuroP were elevated in all diseases, and 4-F_4t_-NeuroP only in Rett syndrome and multiple sclerosis [51]. These results seem to show the relationship of these molecules with these diseases, as has also been shown in this work.

For the F_2t_-dihomo-IsoPs derived from AdA, significant reductions were observed after EXT consumption in all three (17-*epi*-17-F_2t_-dihomo-IsoP, 17-F_2t_-dihomo-IsoP, and *ent*-7(RS)-7-F_2t_-dihomo-IsoP), but, as with 4-F_4t_-NeuroP, this was not observed after PLA consumption. In addition, the 17-*epi*-17-F_2t_-dihomo-IsoP levels presented significant differences, when comparing the study endpoint and in the evolution between groups. It should be noted that the *ent*-7(RS)-7-F_2t_-dihomo-IsoP values showed significant differences, but the cause is unknown.

Elevated values of F_4_-NeuroPs and F_2t_-dihomo-IsoPs are associated with neurodegenerative diseases as well as brain lesions, the former being a biomarker with higher reliability, which requires further investigation to assess whether it could be effective for the early detection of neurological disorders [27]. The F_4_-NeuroPs appear to be much more sensitive markers for oxidative damage to neurons than IsoPs because DHA has a greater sensitivity to oxidation than AA due to the presence of a larger number of methylene double bonds [52]. Meanwhile, the F_2t_-dihomo-IsoPs do appear to be an early marker of lipid peroxidation in Rett syndrome, a disorder that causes developmental delay, especially in areas controlling expressive language and hand movements [26,53].

Libia et al. observed how markers of CNS oxidation in urine samples correlated positively with age [25], which raises the efficacy of promoting polyphenol intake to prevent and not accentuate future problems at the neurological level. The same research group also observed how NeuroP and F_2t_-dihomo-IsoP values in urine samples were increased when comparing high-altitude training versus sea-level training in elite triathletes, which seems to be due to the phenomenon of hypoxia at high altitudes [54]. In another study conducted by the same research group, it was observed that the ingestion of 200 mL of aronia juice in elite triathletes for 45 days reduced markers of CNS lipid peroxidation in urine samples compared to placebo [55]. Consistent with the latter study are data observed in an intervention that evaluated DHA lipid peroxidation and the production of NeuroPs and F_2t_-dihomo-IsoPs in urine samples as markers of oxidative damage in the CNS; improvements were observed after red wine versus ethanol intake, probably due to the ability of melatonin and hydroxytyrosol to cross the blood–brain barrier (BBB) and are highly present in wine [56].

Polyphenols can alter brain function in three compartments: outside the CNS (by improving cerebral blood flow or modulating signaling pathways from peripheral organs to the brain), in the BBB (by altering the input and output mechanisms of several drug-resistant protein-dependent biomolecules), and within the CNS (by directly modifying neuronal and glial cell activity) [57,58,59]. Flavonoids, in particular, can influence the survival cascade and transcription factors by modulating the redox potential of neurons and glia, presenting a protective function against oxidative damage in the brain [60]. In addition, certain polyphenols have the ability to cross the BBB [61,62].

The BBB protects the brain and consists of endothelial cells with tight junctions between them, allowing a permeability in a narrowly selective manner [63]. This barrier modulates the exchange of molecules between the blood and neuronal tissue, regulating the access of nutrients and different compounds to the brain. The BBB is selectively permeable to polyphenols depending on their structural properties [64]. The physiology and state of the brain can be affected by the compounds that manage to cross the BBB, with the microbial metabolites of polyphenols showing higher permeability compared to their original compounds [65]. There are certain flavonoids that are able to cross the BBB [62], such as catechin, quercetin, cyanidin-3-glucoside [66], different anthocyanidins [67], and hesperetin, naringenin, and their derivatives, which all belonging to the subclass of flavones [61]. It is noteworthy that in the product under investigation, Bresciani and collaborators were able to identify quercetin sulfate, myricetin glucuronide, hesperetin sulfate, naringenin glucuronide, and hesperetin glucuronide, among others, in the blood plasma of their subjects [39]. These are compounds that, according to the scientific literature, could cross the BBB.

Polyphenols have a broad spectrum of molecular and cellular actions against neurological degeneration [68]. Brain function is mediated by molecules such as brain-derived neurotrophic factor (BDNF), a neurotrophin that influences the maintenance, survival, growth, and differentiation of neurons, which is more active in brain regions associated with cognition such as the cerebral cortex and hippocampus [69]. Low levels of BDNF are associated with increased neurodegenerative diseases [70]. It has been shown that foods rich in dietary polyphenols have antioxidant and anti-inflammatory activities at the brain level, in addition to being associated with increased BDNF expression [71]. In line with these observations, together with the reduction in NeuroPs and F_2t_-dihomo-IsoPs described in the present investigation, it seems that the EXT used could produce improvements at the brain level by decreasing oxidative damage due to the high polyphenolic content of this product.

The limitations of the study include the fact that a population without previous pathologies was chosen; we believe that the baseline values would have been higher in a pathological population and the results could have been more significant. Another limitation was that the subjects were not exposed to additional OS; it was decided not to do so due to the long intervention period, and to prevent a greater number of losses to follow-up. Another limitation was found in assessing the results of *ent*-7(RS)-7-F_2t_-dihomo-IsoP, where, despite significant reductions after EXT consumption, it was the only variable that began with non-homogeneous values at baseline.

In regard to future clinical trials, it would be appropriate to choose a population with certain previous pathologies, neurological disorders, or with a certain degree of inflammation, such as population with obesity, in order to determine whether there are improvements in these populations. Another possibility could be to subject the population to certain stresses such as physical activity, taking into account that physical exercise itself is capable of improving endogenous antioxidant capacity. On the other hand, stratification by different metabotypes could be carried out to identify, depending on the composition of the microbiota, which types of subjects could benefit more from a particular type of polyphenol.

## 5. Conclusions

The results of the present investigation suggest that the long-term consumption of a nutraceutical with a high polyphenol content is effective in reducing lipid peroxidation of the CNS by reducing different metabolites derived from AdA, which is present in the white matter of the brain, and from DHA, which is present in the gray matter. This may exert a protective effect against oxidative damage in the CNS.

## Figures and Tables

**Figure 1 antioxidants-12-00721-f001:**
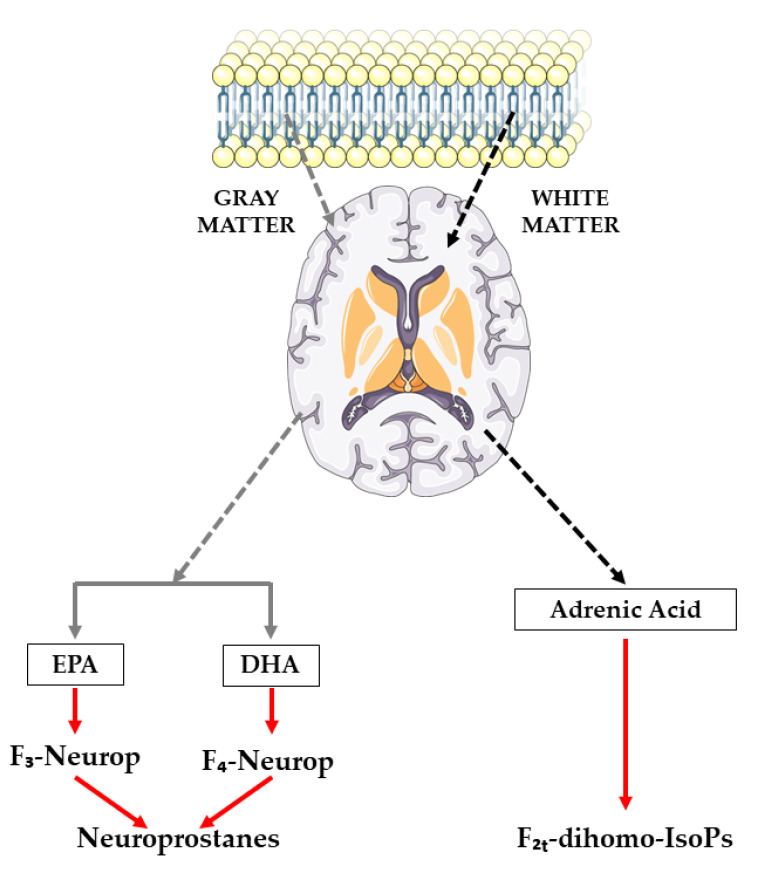
Synthesis route of neuroprostanes and F_2t_-dihomo-isoprostanes.

**Figure 2 antioxidants-12-00721-f002:**
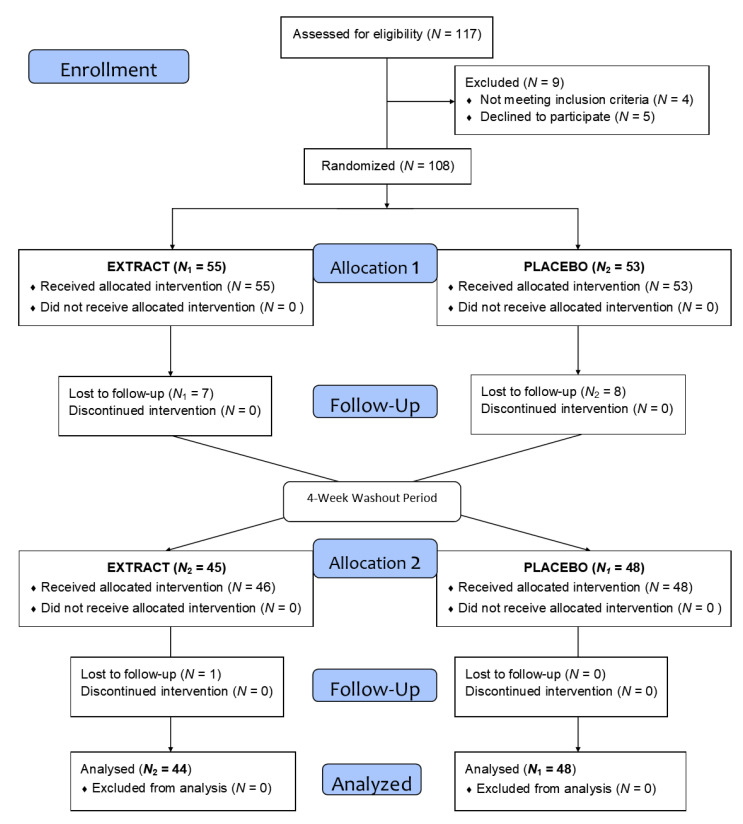
Flow diagram of the intervention.

**Figure 3 antioxidants-12-00721-f003:**
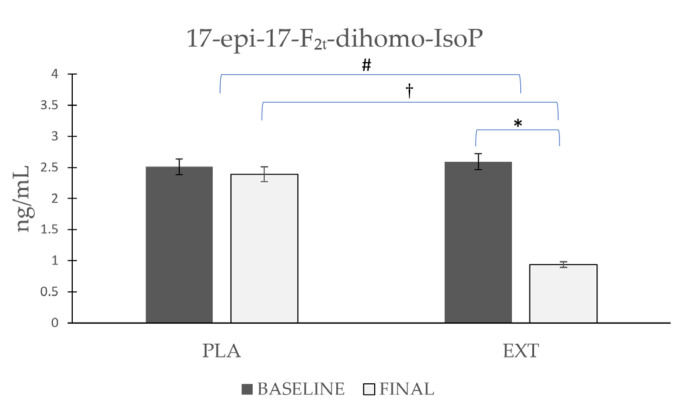
Evolution of 17-*epi*-17-F_2t_-dihomo-IsoP levels in urine during the intervention. *: Statistically significant differences between baseline and final intragroup (*p* < 0.05). ^#^: Statistically significant differences between groups at the end of the intervention (*p* < 0.05). ^†^: Statistically significant differences comparing the evolution between groups during the intervention (*p* < 0.05).

**Figure 4 antioxidants-12-00721-f004:**
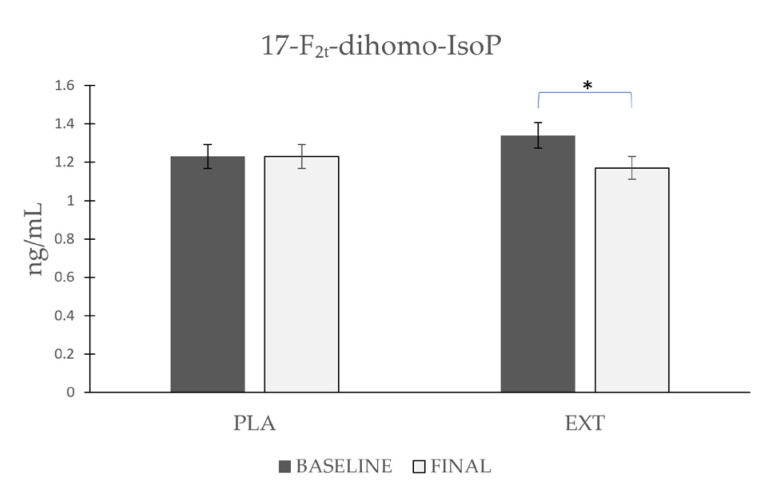
Evolution of 17-F_2t_-dihomo-IsoP levels in urine during the intervention. *: Statistically significant differences between baseline and final intragroup (*p* < 0.05).

**Figure 5 antioxidants-12-00721-f005:**
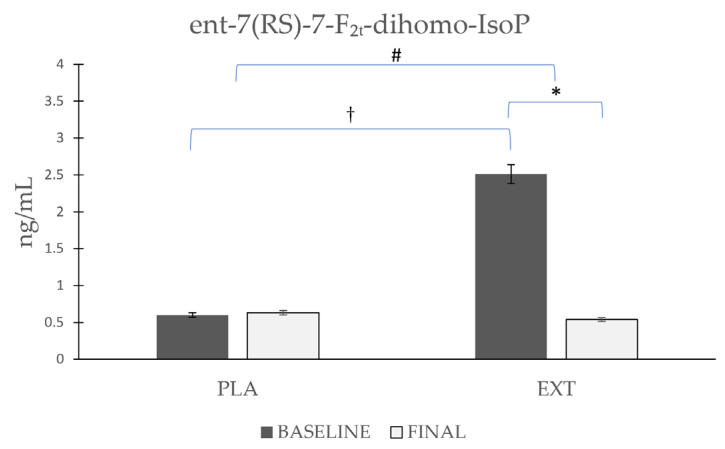
Evolution of *ent*-7(RS)-7-F_2t_-dihomo-IsoP levels in urine during the intervention. *: Statistically significant differences between baseline and final intragroup (*p* < 0.05). ^#^: Statistically significant differences between groups at the end of the intervention (*p* < 0.05). ^†^: Statistically significant differences comparing the evolution between groups during the intervention (*p* < 0.05).

**Figure 6 antioxidants-12-00721-f006:**
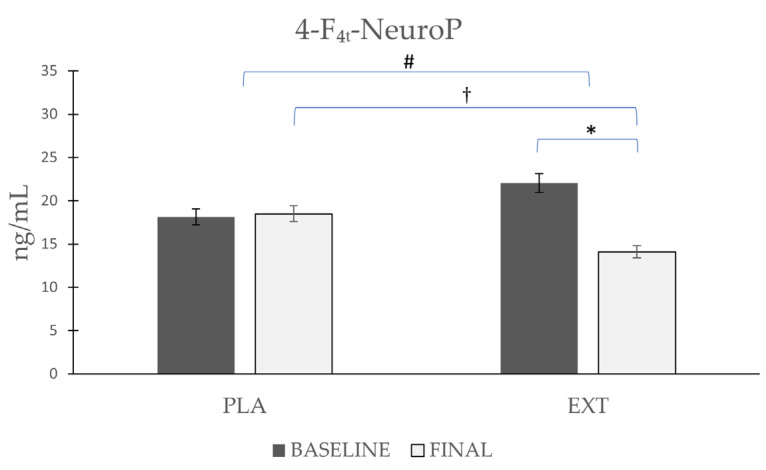
Evolution de 4-F_4t_-NeuroP levels in urine during the intervention. *: Statistically significant differences between baseline and final intragroup (*p* < 0.05). ^#^: Statistically significant differences between groups at the end of the intervention (*p* < 0.05). ^†^: Statistically significant differences comparing the evolution between groups during the intervention (*p* < 0.05).

**Table 1 antioxidants-12-00721-t001:** Neuroprostanes and F_2t_-dihomo-isoprostanes analyzed in the present investigation.

NeuroPs/F_2t_-dihomo-IsoPs	Retention Time (min)	MRM Transition (*m*/*z*)	Molecular Weight (g/mol)
Neuroprostanes derivates from DHA			
4(RS)-4-F_4t_-NeuroP	5.26	377.1 > 271.2	378.5
4-*epi*-4-F_3t_-NeuroP	7.18	379.0 > 219.0	378.5
4-F_4t_-NeuroP *	4.10	377.1 > 333.1	378.5
F_2t_-dihomo-Isoprostanes derivates from AdA			
17-*epi*-17-F_2t_-dihomo-IsoP *	5.90	381.0 > 337.1	382.5
17-F_2t_-dihomo-IsoP *	6.54	381.0 > 337.1	382.5
*Ent*-7(RS)-7F_2t_-dihomo-IsoP *	5.89	381.1 > 363.2	382.5

DHA: docosahexaenoic acid; AdA: adrenic acid; MRM: multiple reaction monitoring; (*) NeuroPs and F_2t_-dihomo-IsoPs that could be quantified in the present investigation.

**Table 2 antioxidants-12-00721-t002:** Demographic data of the study population.

Variable	Total	N_1_	N_2_
N	92	48	44
Men	45	20	25
Women	47	28	19
Age (years)	34 ± 11	33 ± 10	36 ± 12
Weight (kg)	73.10 ± 14.29	70.68 ± 13.88	75.68 ± 14.44
Height (m)	1.72 ± 9	1.71 ± 9	1.73 ± 9
BMI (kg/m^2^)	24.40 ± 3.43	23.87 ± 3.42	24.99 ± 3.38

**Table 3 antioxidants-12-00721-t003:** Oxylipins quantified in the present investigation.

Oxylipins (ng/mL)	Product	Baseline	Final	*p*-1	*p*-2	*p*-3
F_2t_-dihomo-IsoPs						
17-*epi*-17-F_2t_-dihomo-IsoP	Placebo	2.51 ± 0.881	2.39 ± 0.828	0.621	0.001 ^#^	0.001 ^†^
	Extract	2.59 ± 0.853	0.94 ± 0.261	0.001 *		
17-F_2t_-dihomo-IsoP	Placebo	1.23 ± 0.358	1.23 ± 0.406	0.983	0.084	0.572
	Extract	1.34 ± 0.425	1.17 ± 0.308	0.015 *		
*ent*-7(RS)-7-F_2t_-dihomo-IsoP	Placebo	0.60 ± 0.182	0.63 ± 0.181	0.971	0.152	0.001 ^†^
	Extract	2.51 ± 0.643	0.54 ± 0.149	0.001 *		
NeuroPs						
4-F_4t_-NeuroP	Placebo	18.12 ± 6.41	18.50 ± 5.86	0.86	0.02 ^#^	0.025 ^†^
	Extract	22.04 ± 7.47	14.10 ± 5.35	0.001 *		

*: Statistically significant differences between baseline and final (*p* < 0.05). ^#^: Statistically significant differences between groups at the end of the intervention (*p* < 0.05). ^†^: Statistically significant differences comparing the evolution between groups during the intervention (*p* < 0.05). p1 (intragroup), p2 (intergroup), p3 (product * time).

## Data Availability

All data are contained within this article.

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
