# Peer review of "Ability of a Polyphenol-Rich Nutraceutical to Reduce Central Nervous System Lipid Peroxidation by Analysis of Oxylipins in Urine: A Randomized, Double-Blind, Placebo-Controlled Clinical Trial"

_antioxidants, 2023, doi:10.3390/antiox12030721_

Round 1
Reviewer 1 Report
Capability of a polyphenol-rich nutraceutical to reduce lipid peroxidation exclusive to the central nervous system by analysis of Oxylipins in urine: a randomized, double-blind, placebo-controlled clinical trial.
Arcusa R., Carillo J.A., Cerda B., Durand T., Izquierdo A.G., Medina S. et al.
Antioxidants-2141528
The authors describe a reduction of lipid peroxidation products derived from docosatetraenoic acid and docosahexaenoic acid by administering polyphenol supplements in high concentration in a cross-over trial over 2x 16 weeks with a wash-out period of 4 weeks in 92 healthy subjects.
Points of Criticism:
Abstract:
The authors must explain abbreviations at first mention, e.g., EPA, DHA - this is valid for the whole manuscript.
Are there standard reference values for 17-epi-17-F2t-dihomo, 17-F2t-dihomo, ent-7(RS)-7-F2t-dihomo and 4-F4t-Neurop? In addition, the term Isoprostane should be mentioned at least once in this listing.
In line 19, there is a missing space between F2t-dihomo and Isoprostanes (line 19). 4t, was not written in subscript (line 27)? F2t (line 19) standard script vs lines 25, 26 subscript - must be unified throughout the manuscript!
The authors mention a polyphenolic extract - this should be defined in more detail.
Introduction:
For instance, lines 45 and 48 (and others) must be completes at the beginning - the authors must correct this throughout the manuscript.
What do the authors mean by "theremore" (line 52)?
Lines 50-52 are written in Spanish and must be translated into English.
The authors must improve numerous departmental errors (lines 60, 72, etc.), spaces and mistakes at the beginning of sentences (e.g., lines 48, 64, 67, etc.) throughout the manuscript.
Line 72: araquidonic acid should read arachidonic acid
The question arises why the authors make a statement about the TAC although they have yet to measure it. The authors state that this biomarker has been erroneously used in blood plasma (aside from the fact that the present study was performed with urine) without presenting evidence, i.e., a juxtaposition between successful versus unsuccessful studies with this biomarker compared to other antioxidant assays. Subsequently, the authors attempt to establish the misleading use of lipid peroxidation products as a measure of antioxidants and claim to represent the bioavailability of polyphenols (whereas a previous study showed that only 20 out of 92 polyphenols contained in the supplement were detectable in vivo, apart from the antioxidant effect of vitamin C and vitamin E in increased concentrations, among others). Furthermore, these antioxidant effects are said to be specific to neurological diseases, which the authors deduce by measuring these degradation products in those diseases in previous studies (apart from the reports on CVD, diabetes and hypertension). The beneficial effects of antioxidants on lipid peroxidation in various conditions, among others in neurological diseases, have long been known, although healthy subjects participated in the present study - which questions the specificity a priori. The antioxidant effects of polyphenols are a generalised effect on the whole organism (as the authors have recently published in the journal "Molecules" with the same subjects). Thus, it should be noted that the authors measured neither antioxidants nor polyphenols but lipid peroxidation products, which are reduced by these supplements, especially in high concentrations (including vitamin C and vitamin E). This results in numerous contradictions and inconsistencies that must be clarified and presented correctly throughout the manuscript.
The administration of these supplements can be expected to improve the antioxidant capacity, but this depends on various upstream processes, i.e., the initial situation of the subjects (antioxidant supply - or oxidative stress), enzymatic vs non-enzymatic antioxidants, polyphenol content in the blood, polyphenol responders vs non-responders. The isolated measurement of isoprostanes cannot determine all these advance information.
The inappropriate defamation (lines 48-57) is fundamentally unjustified in a scientific publication. The generalised discrediting of biomarkers is incompatible with the high standards of scientific ethics and should be avoided at all costs.
The reference of quotation 7 invokes that the TAC measures a total overview of the antioxidant capacity, e.g., in serum or plasma, including "all" endogenous antioxidants, which includes not only the enzymatic antioxidants but also the potent antioxidants uric acid and bilirubin, which is definitely also the task of an antioxidant assay. The authors of Ref. 7 also point out that there exist TAC methods that are valuable and correlate with polyphenols, which is why they mainly refer to the Folin-Ciocalteu method. A recently published study fully confirms the correlation between polyphenols and antioxidant capacity (Food Bioscience 2022, 49; 101948). Therefore, why do the present paper's authors unspent this specific polyphenol biomarker arises.
The authors should write the caption of Fig.1 in English.
The authors write in the legend of Fig.1 - "Own elaboration" - what does this mean? Are these new findings?
Line 102:
The authors speak of effects on specific CNS lipid peroxidation. This must be put into perspective, as only selected biomarkers were used.
Materials and Methods:
Line 109:
The authors refer to Table 7, although it does not appear throughout the manuscript. This needs to be corrected.
The use of Spanish expressions, including legend Table 1, Table 3 etc., must be translated into English throughout the manuscript.
Line 174: The authors write: "500µL of mM MeOH/HCL mM" - what is meant by this?
Results:
The legend of Fig.2 needs to be included.
Line 216: Did the authors use the Bonferroni test for the posthoc analysis?
Line 262: Why was ent-7(RS)-7F2t-dihomo written in bold?
Line 299 vs 301: The authors use different spellings for the same term - i.e., NeuroPs vs NeuroP - this must be consistent.
Line 314: The authors refer to the association of lipid peroxidation products with neurological diseases, although healthy subjects participated in the present manuscript. This needs to be clarified.
Lines 320-325 should be moved to the weaknesses of the study, which have not been mentioned so far.
The authors point out in Ref. 25 that there is an age-dependent increase in lipid peroxidation - but this is generalised and not isolated to CNS oxidation. This applies in the same way to sport or altitude. Thus, the exclusivity must be relativised by referring to specific areas of the body in addition to mentioning the generalized effects.
Conclusions
Line 371: The effects are not exclusive - therefore, this definition must be removed.
Author Response
Thank you for your comments which helped us to improve the quality of our manuscript. Attached is a word document with all the responses, and an extensive language revision has been requested.

Reviewer 2 Report
The paper entitled "Capability of a Polyphenol-Rich Nutraceutical to Reduce Lipid 2 Peroxidation Exclusive to the Central Nervous System by Anal- 3 ysis of Oxylipins in Urine: A Randomized, Double-Blind, Pla- 4 cebo-Controlled Clinical Trial" is very interesting and original.
I have only some minor comments:
1) Revise grammar in lines 44-55. Some phrases are in Spanish and some letters are cut.
2) Revise some figure titles. They are in spanish.
3) Use EXT instead of EXTRACTO in the figures
4) I miss some future perpespectives at the end of the discussion section
Author Response

(The authors gave the same response as above.)

Reviewer 3 Report
The manuscript entitled “Capability of a Polyphenol-Rich Nutraceutical to Reduce Lipid Peroxidation Exclusive to the Central Nervous System by Analysis of Oxylipins in Urine: A Randomized, Double-Blind, Placebo-Controlled Clinical Trial” submitted to Antioxidants by Dr. Arcusa and co-workers presents the findings of the effects of polyphenol-rich extract on lipid peroxidation. The manuscript/study has serious deficiencies.
The title “Capability of a Polyphenol-Rich Nutraceutical to Reduce Lipid Peroxidation Exclusive to the Central Nervous System….” Is non appropriate in my opinion. EPA and DHA are PUFA ubiquitous and non-exclusive from central nervous system.
Line 44-54. “Among the problems encountered when assessing oxidative stress, the main one is to find validated biomarkers [6]. n addition, to be useful, such markers must show specificity for a given disease, have prognostic value, correlate with disease activity, be reasonably stable, be present in easily accessible tissues, and be cost-effective to measure on a large scale [1]. eral times, total antioxidant capacity in the blood plasma of subjects has been erroneously used. However, such a parameter should be discouraged since it does not provide useful information about the state of the organism [7]. Es conocido como numerosos fitoquímicos se metabolizan fugazmente en moléculas con propiedades biológicas alteradas [7], thermore, according to the scientific literature, it is of greater interest to evaluate the activity of the antioxidative enzymes of the organism [8], which are the ones that bear the main burden of anti-oxidative defense of the organism [9]” This paragraph is a good example of the serious deficiencies along the manuscript. A lot of typographical mistakes, sentences in Spanish mixed in the English text, confused sentences…….Figure 1 mix English and Spanish………..Definitively, the manuscript needs a profound review.
Line 169. “Once urine samples were collected from each subject, they were stored at -80°C until analysis. For enzymatic hydrolysis, samples were allowed to thaw and 1 mL of urine was taken at room temperature to which 100μL of 0.1M acetate buffer, pH 4.9 was added, then 55 μL of enzyme (β-glucuronidase Helix pomatia G-0876) was added to remove glucuronide and sulfate conjugates according to [39,40] and incubated for 2 hours at 37°C in warm bath. After two hours, 500 μL of mM MeOH/HCL mM was added to elicit a protein precipitate, vortexed and centrifuged at 10,000 rpm for 5 min”. I cannot understand the methodology used to oxylipins determination. Why are the samples treated with glucuronidase????
Table 2 can be deleted; this information is also presented in the text.
The findings presented in Table 3 and figures 3-6 are redundant.
Discussion is too much ambiguous. “In addition, certain polyphenols have the ability to cross the BBB [59] In addition, certain polyphenols have the ability to cross the BBB [60] (line 354). What polyphenols? What is the relationship between studied extract????
Line 358. “The brain is mediated by molecules such as brain-derived neu-358 rotrophic factor (BDNF),….” I cannot understand the role of BDNF in this history.
“These data seem to show effectiveness of the extract in terms of prevention of neurological disorders,…”. Neurological disorders are non-studied in this clinical trial, consequently, the main conclusions are not appropriately supported by the findings.
Author Response

(The authors gave the same response as above.)

Round 2
Reviewer 1 Report
If the authors strive to measure oxidative stress, then the TAC is not the first choice. Nevertheless, the authors measured antioxidant effects indirectly through a reduction in oxidative stress parameters by administering highly concentrated polyphenols. The study design thus contradicts the authors' statements in the point-to-point reply. Antioxidant analyses would have significantly increased the manuscript's value in this case. Antecedent, total polyphenols should have been measured. There are also a lot of valid OS methods available to measure diverse biomarkers induced by the action of reactive oxygen species. The authors describe specific parameters in their paper that are highly interesting and valuable for specific questions, as was also noted in the supplement at the end of the manuscript. On the other hand, the authors merely measured iso(neuro)prostanes, which is why a comparison with other methods is not justified. Thus, the authors should focus on their empirically determined parameters, emphasizing their strengths and eliminating vague cross-links with other unmeasured biomarkers, i.e., lines 46-57, from the manuscript.
In Table 1, the changes were not made - Neuroprostanos derivates de DHA and F2t-dihomo-Isoprostanos derivates de AdA still need to be changed. Furthermore, weihgt (Molecular weight) should be corrected. In table 3, Extracto should read Extract.
Author Response
Thank you for your suggestions and comments that help us to improve the quality of the manuscript. We include the responses in word format.
Best regards.

Reviewer 3 Report
Although the revised version of the manuscript entitled “Capability of a Polyphenol-Rich Nutraceutical to Reduce Lipid Peroxidation to the Central Nervous System by Analysis of Oxylipins in Urine: A Randomized, Double-Blind, Placebo-Controlled Clinical Trial” has considered several of the referee comments/suggestions, this last version presents serious deficiencies yet.
I believe that the manuscript explores the effects of polyphenolic extract on urine lipid peroxidation biomarkers related to central nervous system (CNS) but not studies the effects on CNS lipid peroxidation as authors suggested along the manuscript.
A lot of sentences should be reviewed by an English review. I can not understand a great part of the text. Only an example “High oxygen consumption by the brain results in excessive ROS production given that neuronal membranes are rich in PUFA; the fatty acids are most vulnerable to FR attack due to the presence of a higher number of double bonds [33]”
Introduction should analyse the relationship between redox state and arachidonic acid cascade pathways and oxylipins biosynthesis ( doi: 10.1080/10715760000300301; doi: 10.1006/abbi.2001.2439).
Figures, tables and legends should be improved. Figures should be emerged. Spanish and English mix should be corrected (Neuroprostanos derivates de DHA, Extracto…..). “Figure 2. Flow diagram” Flow diagram about??????
Conclusions are not supported by the findings/evidence.
Author Response

(The authors gave the same response as above.)
